# A 'parameiosis' drives depolyploidization and homologous recombination in *Candida albicans*

Matthew Z. Anderson [1,2]*, Gregory J. Thomson[3], Matthew P. Hirakawa[3,4] & Richard J. Bennett [3]*

Meiosis is a conserved tenet of sexual reproduction in eukaryotes, yet this program is see-mingly absent from many extant species. In the human fungal pathogen *Candida albicans*, mating of diploid cells generates tetraploid products that return to the diploid state via a non-meiotic process of depolyploidization known as concerted chromosome loss (CCL). Here, we report that recombination rates are more than three orders of magnitude higher during CCL than during normal mitotic growth. Furthermore, two conserved 'meiosis-specific' factors play central roles in CCL as *SPO11* mediates DNA double-strand break formation while both *SPO11* and *REC8* regulate chromosome stability and promote inter-homolog recombination. Unexpectedly, *SPO11* also promotes DNA repair and recombination during normal mitotic divisions. These results indicate that *C. albicans* CCL represents a 'parameiosis' that blurs the conventional boundaries between mitosis and meiosis. They also reveal parallels with depolyploidization in mammalian cells and provide potential insights into the evolution of meiosis.

[1] Department of Microbiology, The Ohio State University, Columbus, OH 43210, USA. [2] Department of Microbial Infection and Immunity, The Ohio State University, Columbus, OH 43210, USA. [3] Department of Molecular Microbiology and Immunology, Brown University, Providence, RI 02912, USA. [4]Present address: Sandia National Laboratories, Livermore, CA 94550, USA. *email: Anderson.3196@osu.edu; richard_bennett@brown.edu

Sexual reproduction arose early in the eukaryotic lineage and has been maintained by most extant species[1,2]. Sex involves gamete fusion and a doubling of cell ploidy, whereas ploidy reduction is typically achieved via meiosis in which two successive cell divisions halve the DNA content in the cell. Extensive recombination between chromosome homologs is characteristic of meiosis and promotes adaptation through the efficient joining of beneficial alleles and the removal of deleterious ones[3–5]. Meiosis presumably evolved from the apparatus responsible for mitotic cell divisions although its exact origins, as well as the benefits of sex, continue to be debated[6,7]. Additional insight is therefore required into meiosis and the emergence of sex to understand how these fundamental processes arose and have been retained in eukaryotes.

Key aspects of meiosis are conserved at both the cellular and molecular level[8,9]. Typically, segregation of homologous chromosomes occurs during Meiosis I and sister chromatids remain tethered until Meiosis II, which mechanistically resembles a mitotic cell division[10]. Hallmarks of Meiosis I include synapsis of homologous chromosomes followed by extensive recombination, which are both essential for faithful chromosome segregation[11,12]. Central to recombination is the directed formation of DNA double-strand breaks (DSBs) by the topoisomerase-like protein Spo11, which, with the exception of dictyostelids[13], represents a conserved meiosis-specific protein[8,14–17]. Spo11-induced DNA breaks lead to the formation of chromosome crossovers via Holliday junction intermediates or the formation of noncrossovers via synthesis-dependent strand annealing[9,18,19].

Cohesins also play a key role in meiosis including the meiosis-specific kleisin Rec8 that assembles on chromosomes prior to premeiotic DNA replication[20–22]. In the model yeasts *Saccharomyces cerevisiae* and *Schizosaccharomyces pombe*, a protease cleaves cohesin attachments along the chromosome arms during the metaphase to anaphase transition in Meiosis I, and subsequently releases centromeric cohesion during Meiosis II. This mechanism ensures correct segregation of chromosome homologs in the first meiotic division, whereas sister chromatids remain attached until the second meiotic division[23–26].

Both *SPO11* and *REC8* are conserved from yeast to humans and loss of either factor causes defects in synapsis and meiotic recombination[9,27]. However, both genes are also present in species that lack a defined meiosis, which could indicate the presence of uncharacterized sexual cycles or functions unrelated to sexual reproduction[1,28–30]. Moreover, several studies now indicate that 'meiosis-specific' genes can be expressed in tumor cells and are associated with ploidy reduction in polyploid (> 2 N) cells, a process known as depolyploidization[31–33]. There is, therefore, a pressing need to understand how 'meiosis-specific' factors promote ploidy change and genetic exchange during both mitotic and meiotic processes.

Fungal species have contributed extensively to our understanding of ploidy cycling, sexual reproduction, and recombination. Complete sexual cycles exist in many fungi yet atypical ploidy cycles have also been uncovered such as that in the pathogenic yeast *Candida albicans*. Here, efficient mating of diploid **a** and α cells generates tetraploid **a**/α cells, yet neither meiosis nor sexual sporulation has been reported[34–37]. Instead, tetraploid instability can be induced by growth on *S. cerevisiae* 'PRE-SPO' medium, upon which *C. albicans* cells undergo "concerted chromosome loss" (CCL)[38,39]. Mitotic non-disjunction events are thought to drive CCL and yet analysis of a limited number of products suggested that Spo11 may enhance recombination during CCL[39]. However, these studies did not address the degree of homologous recombination accompanying CCL, or the precise role of Spo11 or other meiosis-specific factors during this process.

The studies outlined here reveal the extent of genetic diversity produced via CCL in *C. albicans* and highlight roles for both Spo11 and Rec8 in this process. We demonstrate that Spo11 promotes high levels of DNA double-strand breaks and recombination during CCL, producing recombination frequencies orders of magnitude higher than during normal mitotic growth. Rec8 also enhances recombination frequencies during CCL, and yet Spo11 and Rec8 have opposing effects on chromosome loss as they increase and decrease chromosome stability, respectively. Interestingly, Spo11 also influences chromosome stability and recombination during normal mitotic passaging. We therefore show that two 'meiosis-specific' factors contribute to genome dynamics and genetic exchange during CCL, with additional roles for Spo11 during mitotic growth. These results suggest that *C. albicans* can undergo a 'parameiosis' with potential parallels to depolyploidization processes observed in higher eukaryotes.

## Results

**Recombination and DNA DSB formation during CCL.** To precisely determine the frequency of inter-homolog recombination during CCL, a tetraploid *C. albicans* strain carrying four heterozygous markers (*GAL1*, *URA3*, *HIS1*, and *SAT1*) at different positions on chromosome (Chr) 1 was constructed (Fig. 1a and Supplementary Fig. 1). The genetically marked strain was grown under CCL-inducing conditions (PRE-SPO medium at 37 °C for 7 days) and cells transferred to 2-deoxygalactose (2-DOG) medium to select for *gal1⁻* cells. 2-DOG^R colonies were subsequently evaluated for the presence or absence of *HIS1*, *SAT1*, and *URA3* markers by growth on selective media (Fig. 1a). Using this approach, most 2-DOG^R colonies (70.3%) had undergone a reduction in ploidy and were intermediate between diploid and tetraploid when analyzed by flow cytometry (Fig. 1b and Supplementary Fig. 2), consistent with the frequent formation of aneuploid CCL progeny[38,39].

Growth under CCL-inducing conditions increased *GAL1* loss in tetraploid cells by ~ 400-fold relative to culture on standard yeast extract peptone dextrose (YPD) medium at 30 °C (Fig. 1c). Furthermore, recombination frequencies within the *HIS1-SAT1* and *SAT1-URA3* genetic intervals in 2-DOG^R progeny were ~ 2700-fold and ~ 8200-fold higher, respectively, when grown under CCL-inducing conditions compared with standard YPD growth (Fig. 1d, e). This corresponds to 1 in 10 cells, on average, having undergone recombination within the tested Chr1 intervals during CCL, indicative of high recombination frequencies.

We next examined whether DSB formation, a hallmark of a conventional meiosis, is observed during CCL. To test this, we introduced a GFP-tagged version of Gam protein from bacteriophage Mu into tetraploid *C. albicans* cells. Gam-GFP has been shown to bind to DSBs and can be used as a visual readout of DSB formation in live cells[40]. In control experiments, doxycycline-induced Gam-GFP expression generated a nuclear GFP signal in *C. albicans* cells treated with ionizing irradiation (Supplementary Fig. 3 and 4). A subset of tetraploid *C. albicans* cells grown under CCL-inducing conditions also displayed an intense Gam-GFP signal within the nucleus (Fig. 1f, Supplementary Fig. 5), whereas tetraploid cells cultured under standard growth conditions did not. Growth under CCL-inducing conditions similarly promoted the formation of DSBs based on the prevalence of cells positive for terminal deoxynucleotidyl transferase (TdT) dUTP Nick-End Labeling (TUNEL) staining (Supplementary Fig. 6). Together, these results establish that there is an increased prevalence of DSBs during CCL relative to normal mitotic growth.

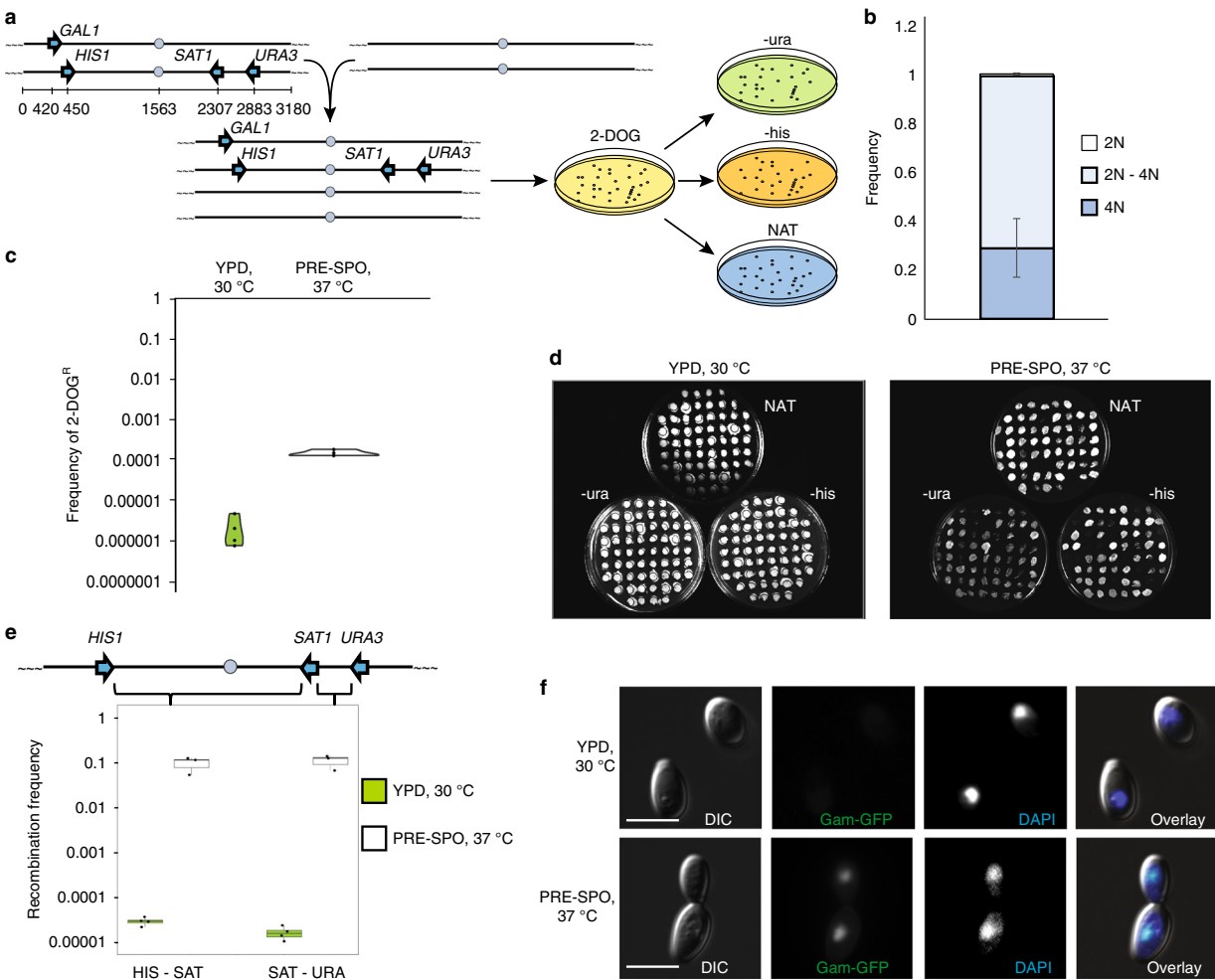

**Fig. 1** High rates of inter-homolog recombination during concerted chromosome loss (CCL). **a** Schematic showing the construction and screening of a genetically marked tetraploid strain for monitoring chromosome loss and recombination. **b** Frequency of different ploidy states in 2-DOG$^R$ ($gal1^-$) progeny derived from tetraploid cells by CCL. Error bars signify standard deviation. $n = 3$ biologically independent experiments. **c** Frequency of 2-DOG$^R$ colonies following growth of the genetically marked tetraploid strain on solid agar under standard conditions (green; YPD, 30 °C) or under conditions that promote CCL (white; PRE-SPO medium, 37 °C) for 7 days. $n = 4$ biologically independent experiments. **d** Example images of selection plates indicating growth of progeny on selective media for different markers. **e** Recombination frequencies in *HIS1-SAT1* and *SAT1-URA3* intervals following growth on standard medium (green) or CCL-inducing medium (white) for 7 days. *SAT1* encodes resistance to nourseothricin (NAT$^R$). Boxplots are presented as the 75th to 25th percentile with the thick line denoting the median. Whiskers indicate the largest and smallest values within 1.5 × of the interquartile range. **f** *C. albicans* tetraploid cells expressing the Gam-GFP construct following 24 h growth on PRE-SPO or YPD medium (CCL-inducing or standard growth conditions, respectively). Scale bar, 5 μm. Error bars indicate standard deviation

**Meiosis-specific factors regulate parasexual processes.** To evaluate the functions of 'meiosis-specific' genes in CCL, we focused on *C. albicans SPO11* and *REC8* homologs, given that these genes play critical and conserved roles in meiosis from yeast to mammals. Both genes were individually deleted in tetraploid *C. albicans* strains carrying genetic markers along Chr1 (Supplementary Fig. 1). Interestingly, deletion of *SPO11* led to ~ 10-fold higher frequencies of 2-DOG$^R$ colonies reflecting increased *GAL1* marker loss relative to the wildtype control (two-sample $t$ $(6.12) = -4.31$, $p = 1.57E-3$, Fig. 2a, b). Complementation of the Δ*spo11* mutant with a single copy of *SPO11* was sufficient to restore normal levels of marker stability during CCL (Δ + *SPO11* (WT) vs. WT: two-sample $t$ $(9.49) = 0.73$, $p = 0.48$). In contrast, integration of a mutated *SPO11* allele lacking the putative catalytic residue (Y65F substitution, Supplementary Fig. 7) failed to rescue the mutant phenotype (Δ + *SPO11*(YF) vs. WT: two-sample $t$ $(5.13) = -3.75$, $p = 5.25E-3$, Fig. 2a, b). Flow cytometry measurements of DNA content in cells 48 h after inducing CCL revealed that Δ*spo11* cells generally had ploidy levels that were

lower than either the wildtype or *SPO11*-complemented strains (Supplementary Fig. 2). A slight, but not significant, reduction in DNA content was observed in CCL progeny from *SPO11*-complemented cells relative to those from wildtype cells. Overall, these results indicate that Spo11 promotes chromosome stability (or reduces chromosome loss) under CCL conditions.

Deletion of *SPO11* also led to marked changes in recombination frequencies during CCL. Thus, tetraploid strains lacking *SPO11* showed a ~ 24-fold reduction in recombination rates in the *HIS1-URA3* interval on Chr1 compared with the wildtype strain (Wilcoxon test (W(40)), $p = 5.04E-5$, Fig. 2c, d, and Supplementary Fig. 8). Reintegration of a wildtype *SPO11* allele into the Δ*spo11* tetraploid strain restored recombination frequencies (Δ + *SPO11*(WT) vs. WT: Wilcoxon test (W(26)), $p = 0.29$, Fig. 2c, d). In contrast, complementation with the *SPO11*-YF allele failed to complement the recombination defect during CCL (Δ + *SPO11*(YF) vs WT: Wilcoxon test (W(34)), $p = 1.75E-4$, Fig. 2c, d). A Gam-GFP signal was present only at background levels in most Δ*spo11* cells during CCL indicative of

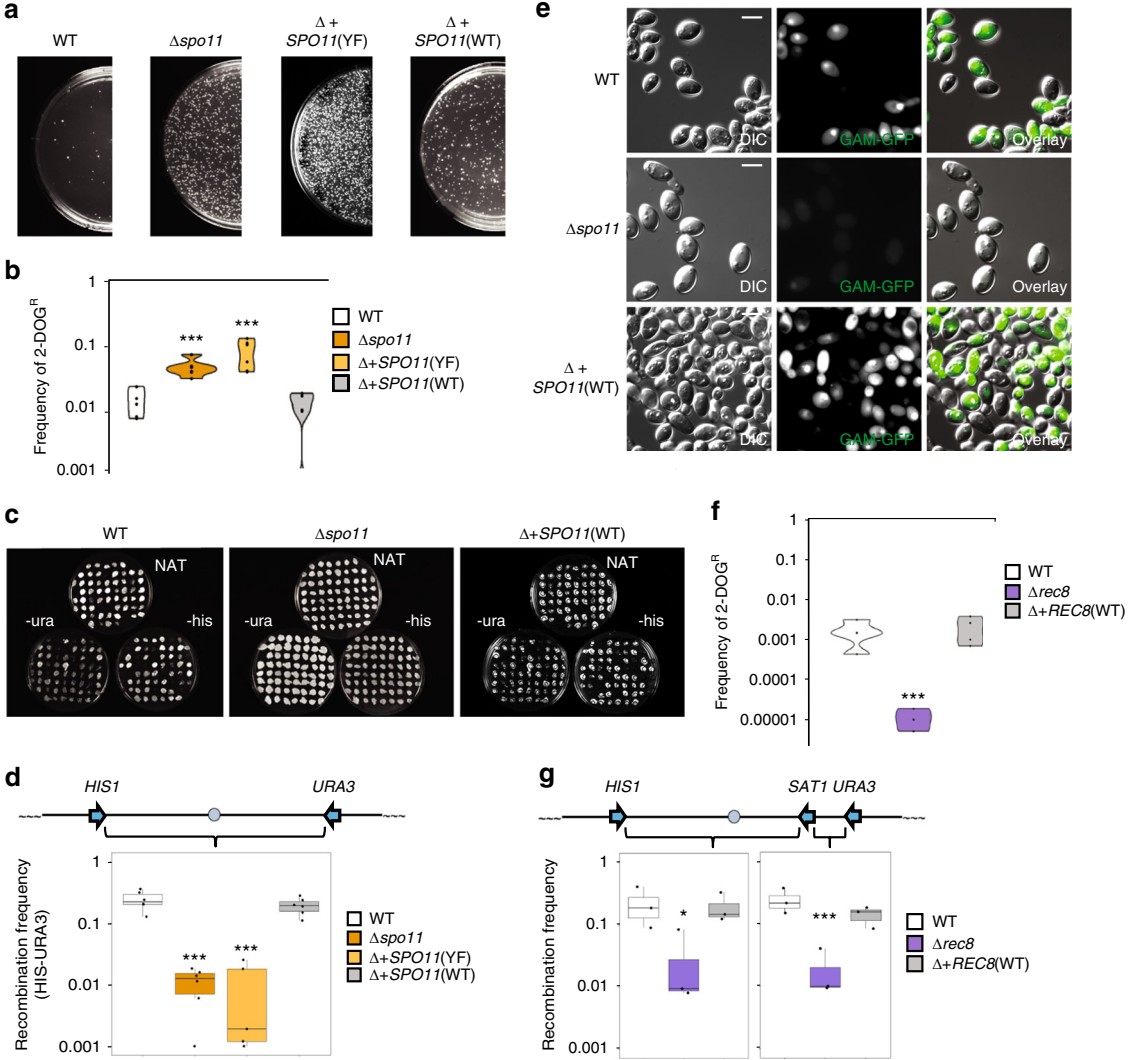

**Fig. 2** Homologous recombination during CCL requires the 'meiosis' factors Spo11 and Rec8. **a** Images of plates illustrating 2-DOG^R colonies produced by CCL in *C. albicans* wildtype and Δ*spo11* tetraploid strains, as well as in Δ*spo11* strains complemented with wildtype *SPO11*, *SPO11*(WT), or an active site mutant, *SPO11*(YF). **b** Quantification of the frequency of 2-DOG^R colonies in different strain backgrounds following CCL. n = 6 biologically independent experiments. **c** Images of plates examining CCL progeny grown on different selective media. **d** Frequency of recombination in CCL progeny from different strain backgrounds. n = 6 biologically independent experiments. **e** Analysis of Gam-GFP signal in tetraploid strains with different *SPO11* genotypes undergoing CCL. Images shown after 24 h on PRE-SPO medium with 50 μg/mL doxycycline at 37 °C. Scale bars = 5 μm. **f** Quantification of the frequency of 2-DOG^R colonies in different *REC8* strain backgrounds following CCL. n = 4 biologically independent experiments. **g** Frequency of recombination in CCL progeny derived from different *REC8* strain backgrounds in *HIS1-SAT1* and *SAT1-URA3* intervals. n = 3 biologically independent experiments. * denotes p < 0.05. *** denotes p < 0.001 as calculated by two-sample t **b**, **f** or Wilcoxon **d**, **g** tests. Boxplots are presented as the 25th to 75th percentile with the thick line denoting the median. Whiskers indicate the largest and smallest values within 1.5 × of the interquartile range. White, orange, light orange, and gray denote WT, Δ*spo11*, Δ+*SPO11*(YF), and Δ+*SPO11*(WT), respectively. White, purple, and gray denote WT, Δ*rec8*, and Δ+*REC8*(WT), respectively

reduced DSB formation, and this signal was restored in the WT *SPO11*-complemented strain (Δ + *SPO11*(WT) vs. Δ*spo11*: two-sample *t* (1.12) = 9.59, *p* = 0.015; Fig. 2e and Supplementary Fig. 5). Examination of cells complemented with the *SPO11*-YF allele revealed fewer cells were Gam-GFP positive than those with the WT allele, but significantly more cells were positive than in Δ*spo11* cells (Supplementary Fig. 5). This suggests that the mutant Spo11 protein may contribute to DSB levels via a non-enzymatic mechanism. Together, these experiments establish that Spo11 promotes DSB formation and supports both chromosome stability and DNA recombination in polyploid cells undergoing CCL.

Next, we evaluated the role of a second meiosis-specific factor, *REC8*, in tetraploid cells. Interestingly, loss of *REC8* had the

opposite effect on chromosome stability during CCL to that of *SPO11*, as Δ*rec8* mutants showed a 48-fold reduction in the formation of 2-DOG^R colonies relative to the wildtype control (two-sample *t* (6) = 3.73, *p* = 9.83E-3, Fig. 2f, Supplementary Fig. 9a and 10). Similar to loss of *SPO11*, however, deletion of *REC8* reduced the recombination frequency (~ 12-fold) in the *HIS1-SAT1* and *SAT1-URA3* intervals compared with the wild-type strain (Fig. 2g, Supplementary Fig. 9b). Recombination frequencies were restored to wildtype levels by reintroduction of one copy of *REC8* into the Δ*rec8* background (Fig. 2g). Both *SPO11* and *REC8* therefore promote high levels of recombination in polyploid cells undergoing CCL. However, *SPO11* enhances chromosome stability during CCL whereas *REC8* has a destabilizing effect on the genome under these conditions.

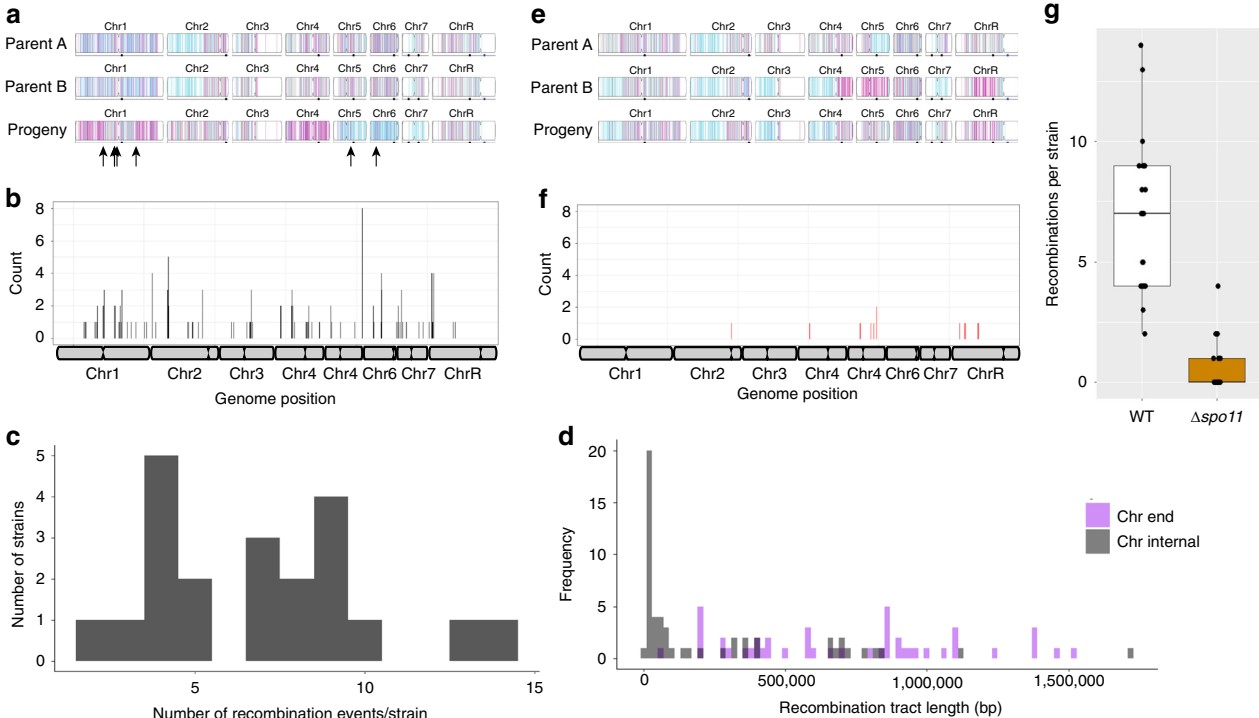

**Fig. 3** Genomic analysis of *C. albicans* CCL progeny. Parental and progeny genotypes are shown for three representative strains in WT **a** and Δ*spo11* **e** backgrounds. The parental genotypes are indicated on the top two lines with a representative progeny underneath. Arrows indicate recombination breakpoints. Homozygous alleles for homolog A or B are represented by cyan and magenta lines, respectively, with heterozygous positions denoted as gray. The recombination breakpoints were mapped to the genome across all 21 genotyped WT **b** and Δ*spo11* **f** progeny. **c** The number of recombination events within each WT progeny following CCL. **d** The length of each recombination tract in WT progeny was determined for chromosome internal tracts (gray) and tracts extending to the chromosome end (purple). *n* = 46 biologically independent data points. **g** The number of recombination breakpoints was compared for WT (white) and Δ*spo11* (orange) progeny. Boxplots are presented as the 25th to 75th percentile with the thick line denoting the median. Whiskers indicate the largest and smallest values within 1.5 × of the interquartile range. *n* = 20 biologically independent samples

**Genome-wide patterns of recombination following CCL.** To investigate global patterns of homologous recombination, we genotyped CCL progeny using double digest restriction-site associated DNA sequencing (ddRAD-Seq), which selectively sequences a large number of marker positions across the genome (see Methods). Wildtype tetraploid cells were induced to undergo CCL and 21 2-DOG$^R$ progeny analyzed by ddRAD-Seq. Approximately half (10/21) of these CCL isolates were aneuploid, with 13.6% (20/147) of the chromosomes present as supernumerary copies. Trisomies of several smaller chromosomes (Chrs 4, 5, and 6) were the most common in CCL progeny (Supplementary Fig. 11a). Interrogation of ddRAD-Seq data revealed multiple recombination events in the 21 progeny (Fig. 3a, b). The number of detectable recombination events varied between 1 and 14 crossovers per strain and recombinant segments ranged from ~10 kb to 1.5 Mb (Fig. 3c). Recombination tracts extending to the ends of chromosome arms could represent break-induced replication (BIR) or crossover events, and these tracts were significantly longer than internal recombination tracts (Wilcoxon test (W(2114), *p* = 1.66E-7, Fig. 3d)). Recombination frequency correlated with chromosome size (Pearson, *r* = 0.78, *p* = 0.027, Supplementary Fig. 12) although breakpoints were not evenly distributed across all chromosomes (Fig. 3b). Mapping the 149 recombination events within the 21 progeny highlighted regions containing multiple recombination events near the centromere of Chr3, the left arm of Chr2, and telomere-proximal regions on ChrR and Chr4. These regions spanned 20–40 kb and represent potential hotspots for recombination, but were not obviously enriched for repetitive genomic elements that are associated with increased recombination frequencies[41].

We also used ddRAD-Seq to examine global recombination patterns in 21 CCL progeny from the Δ*spo11* tetraploid and found that recombination frequencies were significantly lower in Δ*spo11* progeny than in wildtype progeny (Mann–Whitney *U* test (W(456)), *p* = 2.23E-8, Fig. 3g). Only 13 recombination events were detected in the 21 Δ*spo11* progeny, with most progeny (15 out of 21) having no detectable recombination events (Fig. 3e, f). Recombination tracts that did not extend to the chromosome ends were, on average, sixfold shorter in Δ*spo11* cells than in wildtype cells (47 kb vs 260 kb, respectively), whereas those extending to the telomere were similar in size between backgrounds (535 kb vs 734 kb, respectively). This suggests that Spo11 selectively impacts the tract length of internal recombination events, although there was insufficient power to compare with the wildtype owing to the limited number of events in the mutant strain. We also note that only 4 of the 21 Δ*spo11* progeny were aneuploid owing to five imbalanced chromosomes (3.4% of 147 chromosomes; Supplementary Fig. 11b), lending support to the model that chromosome instability (and a return to the 2 N state) may be enhanced by *SPO11* loss.

**A functional role for *SPO11* during mitotic growth.** In addition to evaluating *SPO11* and *REC8* function during CCL, we examined their potential role(s) in genome stability and recombination during normal mitotic growth in diploid and tetraploid cells. Although *SPO11* and *REC8* are both expressed in mitotic cells[39,42], a functional role for these genes during mitotic divisions has not been reported. Genetically marked diploid or tetraploid strains were passaged in YPD medium at 30 °C or 37 °C (serially

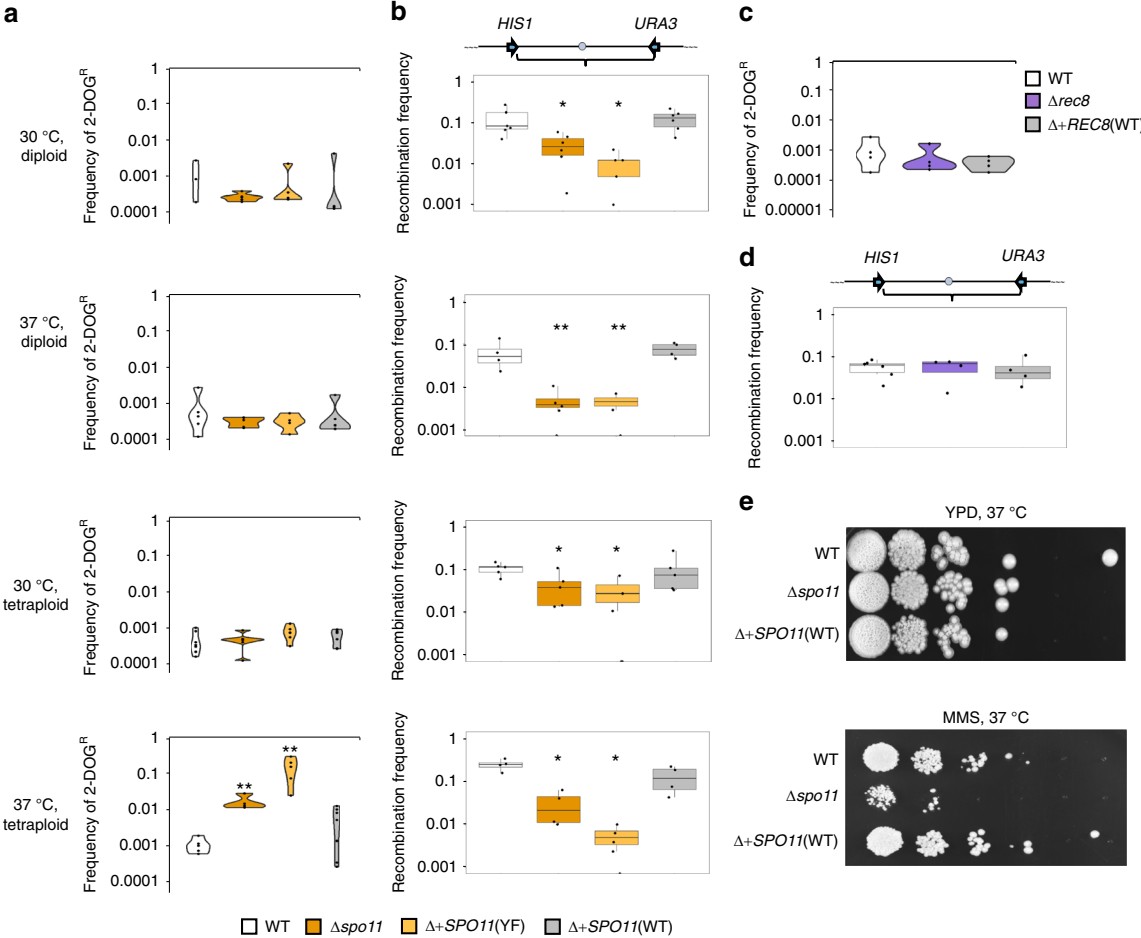

**Fig. 4** Spo11, but not Rec8, promotes recombination in mitotically dividing *C. albicans* cells. **a–d** Genetically marked diploid and tetraploid *C. albicans* strains were serially passaged in YPD medium at 30 °C or 37 °C (cells diluted 1:100 every 24 h), and evaluated for loss of the *GAL1* marker (2-DOG$^R$ frequency) and for the frequency of recombination in the *HIS1-URA3* interval. **a** Frequency of 2-DOG$^R$ colonies in diploid/tetraploid wildtype and *SPO11* mutant strains. $n = 4$, 5, 6, and 6 biologically independent experiments for 30 °C diploid, 37 °C diploid, 30 °C tetraploid, and 37 °C tetraploid, respectively. **b** Frequency of recombination between *HIS1* and *URA3* on Chr1 in diploid/tetraploid wildtype and *SPO11* mutant strains. $n = 4$, 4, 5, and 5 biologically independent experiments for 30 °C diploid, 37 °C diploid, 30 °C tetraploid, and 37 °C tetraploid, respectively. **c** Frequency of 2-DOG$^R$ colonies in wildtype or *REC8* mutant tetraploid strains passaged at 30 °C. $n = 4$ biologically independent experiments. **d** Analysis of recombination in the *HIS1-URA3* interval for wildtype or *REC8* mutant tetraploid strains passaged at 30 °C. $n = 4$ biologically independent experiments. **e** Tenfold spot dilution assays performed for wildtype, Δ*spo11*, and *SPO11*-complemented tetraploid cells grown on YPD medium in the presence or absence of 0.01% methyl methanesulfonate (MMS) and imaged after 2 days at 37 °C. * denotes $p < 0.05$. ** denotes $p < 0.01$ as calculated by two-sample $t$ (chromosome loss) or Wilcoxon (recombination frequency) tests. Boxplots are presented as the 25th to 75th percentile with the thick line denoting the median. Whiskers indicate the largest and smallest values within 1.5 × of the interquartile range. White, orange, light orange, and gray denote WT, Δ*spo11*, Δ+*SPO11*(YF), and Δ+*SPO11*(WT), respectively. White, purple, and gray denote WT, Δ*rec8*, and Δ+*REC8*(WT), respectively

diluted every 24 h for 7 days) and analyzed for the presence/absence of markers on Chr1 (Fig. 1a). In general, similar levels of *GAL1* marker stability were observed in diploid and tetraploid cells grown at both 30 °C and 37 °C (Fig. 4a, Supplementary Fig. 13a). However, recombination frequencies varied markedly between conditions, with tetraploid cells exhibiting higher levels of recombination at 37 °C than at 30 °C, as well as higher recombination frequencies than diploid cells grown at either temperature (Fig. 4b, Supplementary Fig. 13b). Notably, loss of *SPO11* activity resulted in a decrease in recombination in diploid and tetraploid cells at both temperatures, as recombination frequencies were lower in Δ*spo11* mutants (2–13-fold) and in cells expressing the *SPO11-YF* allele (5–54-fold) than in wildtype controls (Fig. 4b, Supplementary Fig. 13b). Reintegration of an intact *SPO11* allele complemented the recombinational defect of the Δ*spo11* mutant in mitotic cells (Fig. 4b, Supplementary Fig. 13b).

We also note that *SPO11* influenced the stability of the *GAL1* marker during mitotic passaging, as tetraploid Δ*spo11* cells grown at 37 °C experienced a 16-fold increase in *GAL1* marker loss relative to wildtype cells (two-sample $t$ (3.69) = 5.50, $p$ = 1.86E-3), whereas cells expressing the *SPO11-YF* variant showed a 135-fold increase in *GAL1* loss compared with wildtype cells (two-sample $t$ (3.01) = 6.53, $p$ = 0.013) (Fig. 4a, Supplementary Fig. 13a and 14). In contrast to *SPO11*, cells lacking *REC8* displayed no change in recombination frequency or *GAL1* marker stability during standard mitotic growth (Fig. 4c, d).

To determine whether Spo11 also contributes to mitotic DNA repair phenotypes in response to genotoxic stress, we compared wildtype and Δ*spo11* diploid cells in the presence of the DNA damaging agent methyl methanesulfonate (MMS). We found that diploid Δ*spo11* cells showed a reproducible increase in MMS susceptibility relative to wildtype cells indicating a defect in DNA repair (Fig. 4e), whereas exposure to other cellular stresses did not

produce detectable $\Delta spo11$ phenotypes (Supplementary Fig. 15). These findings establish that Spo11 can function in DNA recombination and repair in mitotically dividing *C. albicans* cells, in addition to its more specialized role during CCL.

## Discussion

The current study details key aspects of CCL in *C. albicans*, during which tetraploid cells undergo extensive genome instability leading to a reduction in ploidy and experience high levels of homologous recombination. Given the apparent lack of a conventional meiosis, this mechanism provides an effective pathway by which *C. albicans* cells can complete a diploid–tetraploid–diploid parasexual cycle and generate diverse progeny. Understanding of the CCL process can therefore shed light on alternative mechanisms of sexual/parasexual reproduction, as well as detailing how parasex produces genetic diversity in this important human pathogen.

Our experiments establish that high recombination frequencies accompany CCL in *C. albicans*, with inter-homolog exchanges on Chr1 being more than three orders of magnitude higher than those during normal mitotic passaging. We also reveal prominent roles for the 'meiosis-specific' factors Spo11 and Rec8 in regulating genome dynamics during CCL, with parallels to their conventional roles during meiosis. In particular, it is now evident that Spo11 introduces extensive DSBs during both CCL and meiosis, and that Spo11 and Rec8 are similarly required for promoting high levels of inter-homolog recombination during both of these processes. We therefore propose that CCL represents a 'parameiosis'[43] in which both ploidy reduction and homologous recombination are dependent on key meiosis-specific factors. CCL clearly differs from a conventional fungal meiosis, however, in that it is highly uncoordinated and ploidy reduction is uncoupled from the process of sexual sporulation, as the latter has never been observed in *C. albicans*.

In meiosis, Spo11 function can affect chromosome pairing at two steps: first, Spo11 promotes homolog alignment independent of its ability to form DSBs, and second, a more precise pairing of homologs occurs during synapsis that is dependent on Spo11-mediated DSBs[44–46]. We found that chromosome instability in *C. albicans* cells undergoing CCL was impacted by Spo11 catalytic activity, as mutation of the active site increased genome instability, which may mirror its role in promoting chromosome pairing during meiosis. Furthermore, loss of Spo11 clearly diminished, but did not abrogate, DSB formation and recombination during CCL, indicating that additional factors contribute to these processes. Surprisingly, Spo11 also impacted DNA repair and recombination during mitotic growth of *C. albicans* diploid and tetraploid cells, revealing a general role for this factor in genome dynamics beyond its function in CCL.

In marked contrast to Spo11, *C. albicans* tetraploid cells lacking Rec8 exhibited reduced *GAL1* marker loss during CCL. The cause of this phenotype and its relationship to the meiotic function of Rec8 is unclear, although Spo11 and Rec8 have opposing roles in promoting S-phase progression in *S. cerevisiae* meiosis[45]. Interestingly, recent experiments also revealed that ectopic over-expression of *REC8* can increase chromosome loss in mitotic *S. cerevisiae* cells[47]. This result is in line with our current findings where Rec8 enhances chromosome instability in *C. albicans* cells, at least under CCL conditions.

It is possible that parameiosis represents an ancestral mechanism for ploidy cycling and thus reflects an intermediary step in the evolutionary path from mitosis to meiosis. The dual function of *C. albicans* Spo11 in both mitotic and parameiotic recombination could also indicate a general role in DNA repair prior to being co-opted for meiosis-specific purposes. However,

given the ubiquity of meiosis in the eukaryotic kingdom[48] and its presence in other extant *Candida* species[48–50], a more parsimonious model is that the ancestor of *C. albicans* underwent meiosis and that this program then degenerated into the parameiotic program present today. In line with this, certain *C. albicans* genes (e.g., *DLH1/DMC1*) are able to complement for the loss of their meiotic orthologs in *S. cerevisiae*[51], whereas others (e.g., *NDT80*) have been reprogrammed for alternative functions such as biofilm formation[52].

Regardless of whether parameiosis is an ancestral program or more recently derived, this study highlights how mating and ploidy cycling could have evolved without the precise chromosome dynamics that characterize a conventional meiosis. Moreover, despite its rudimentary mechanism, CCL can still achieve several key functions of meiosis including (1) a reduction in ploidy, (2) generation of recombinant progeny, and (3) the purging of deleterious alleles/unmasking of advantageous recessive alleles[43]. The uncoupling of ploidy cycling from sexual sporulation may have been a key step in the emergence of parameiosis in *C. albicans* and conferred a fitness advantage to a commensal organism whose sexual spores would likely be antigenic to the host[53].

Finally, we note that non-meiotic processes are now recognized as important drivers of ploidy change in multiple eukaryotic species. For example, the filamentous fungus *Aspergillus nidulans* can undergo a parasexual cycle in which diploid cells readily lose chromosomes to return to the haploid state, a process that can generate population diversity de novo[54,55]. Parasexuality has also been described in the parasitic protozoans *Trypanosoma cruzi* and *Giardia intestinalis*, in which genetic exchange can take place in the absence of meiosis but where 'meiosis-specific' genes such as *SPO11* are expressed, at least in *G. intestinalis*[56–58]. Mechanistic parallels also extend to depolyploidization processes observed in mammalian cells. In particular, polyploid tumor cells can undergo depolyploidization that is accompanied by upregulation of *SPO11* and *REC8* genes[32,33]. Expression of such genes in several cancer types is thought to promote genome instability and drive the formation of aneuploid products[31,59]. These findings are analogous to the high rate of aneuploid formation arising in *C. albicans* cells as a result of CCL. Thus, diverse eukaryotic species appear to be able to use conserved meiosis genes for alternative mechanisms driving ploidy reduction. These studies establish that new roles for meiosis-specific factors are being uncovered that blur the boundaries between conventional mitotic and meiotic programs, as well as providing potential insights into how ploidy cycling and meiosis could have evolved in early eukaryotes.

## Methods

**Media used**. Media was prepared by dissolving the appropriate nutrient into ddH$_2$O and autoclaving as previously described[60,61]. YPD plates containing 200 μg/mL nourseothricin (NAT) were used for selection of strains that were resistant to nourseothricin (SAT$^R$ strains)[62]. *S. cerevisiae* pre-sporulation ('PRE-SPO') medium contained 0.8% yeast extract, 0.3% peptone, 10% glucose (added prior to autoclaving), and 2% agar[63]. Medium with 2-deoxygalactose (2-DOG) consisted of Synthetic Complete medium supplemented with 2% glycerol, 2 mg/mL 2-DOG (Fluka #31050), and 2% agar. Maltose plates were prepared as YP media supplemented with 2% maltose and 2% agar.

**Construction of genetically marked strains**. All strains are listed in Table S1. The genetically marked chromosome strain was produced by multiple rounds of targeted integration in *C. albicans* strain SN78 obtained from the Noble lab (UCSF)[64]. To target *GAL1*, a plasmid was constructed in which the 5′ sequence flanking *GAL1* was PCR amplified with primers gcggccatgggcccTCCCGACACCAAAATCA-TAATT and caggcgcgCTCGAGTCAAACGTAGGAACTGACATGG and the 3′ flank was PCR amplified with primers gcggccaCCGCGGCCATCTATGGG-TAGTTGTATTG and caggcgcgGAGCTCGACTTGTATAAGCCTACTTTGC. These products were cut with enzymes *Apa*I/*Xho*I and *Sac*II/*Sac*I, respectively, and cloned into pSFS2A[62]. The resulting plasmid, pRB472, was treated with *Apa*I and *Sac*I and the products used to delete *GAL1* by transformation into SN78 followed

by recycling of the *SAT1* selectable marker by induction of FLP by plating to maltose. The allele deleted was phased to a specific chromosome homolog by sequencing of the remaining homolog. A single copy of *HIS1* was amplified from the genome using primers ggcgccAAGCTTGGTTGGCTCTCTTAATACGAT and gccggcGGATCCTGGTAAACCTGAGTGAGATG that target HIS1 adjacent to SNP #23 in the genome[65]. Phasing of *HIS1* relative to SNP #23 was performed using primers AGCCAACCATATTTCAGGATTGAC and GTGCCAACTAGTA ATGGTTGTCAT and digest with *Hinc*II. Following *HIS1* phasing, *URA3* was amplified from the SC5314 genome using primers CAGAAAAAAAAATTTTGAT GATGAGATGGTCTATTTATAGTGCGCGAAATTTAGAAGCATCTCATGGT TTTCTAGAAGGACCACCTTTGA and ATGAAAATTTATTTGATGTCTTGGG AGCCCTAATTATTTTTCTTCTTCATTAATCTTTGATCTCATCGCCGTTGCT GTAGTGCCATTGAT and introduced at SNP #42[66]. Phasing of *URA3* was performed using primers GAAATCCACCGCATAAGAAATGGTT and CTGGTGAG GCATGAGTTTCT and digest with *Mlu*I. *MTL***a** or *MTL*α were deleted using the previously described plasmids pRB101 and pRB102, respectively[67]. Introduction of *SAT1* at SNP #30 is accomplished by amplifying the *SAT1* marker from pSFS2a[62] with primers AGTCATTAACTTGAAACTTCACCTTCCACTACAATCCTTTTC TCAAAGCAAAGCTAAAAAGTAATGCTACCCATCATAAAATGTCGAGCG and TAAGATTGATGTGTAATGGGAACGTTATCAAAGTAAATAAGTGTGGT GCGACAAACGACAGCCAGAAACGCTCTAGAACTAGTGGATC. Phasing of *SAT1* as SNP #30 was determined using primers ACAGCGATGTACTGGTA CTG and GCATGGTTTGAACAAGTGATAGAGT and digest with *Alu*I.

Deletion of *SPO11* and *REC8* (orf19.3589 and orf19.776, respectively) was performed by homologous recombination of PCR products amplified from pSFS2A[62] with 70 bp of flanking sequence to the desired locus. Two successive rounds of PCR was performed to target each of the two *SPO11* alleles with CTATAACAATATCTCATACTGAGTCCATGTATGTTGGAAACAATCATATC ATCAATTTTCTGAAAACGGGAGGGAACAAAAGCTGGAGCT/ CCATCTT TTTCAATGTAGACTTTGTAGTCTAAAGTGTTGGCAGCACCAACTTGACGA GTAGAGTATGAAGACATCCTCGAGGAAGTTCCTATACT for deletion of the first allele and CTGAAAACGGGAAGACTAACTTTCTTTAAGAAACAAATTA CAGGTGCTCCTAAATATTTGGGAACAAGTTGGGAACAAAAGCTGGAG CT/ TTTGGGAACTTTTGCTCGATTTTGTTGAGAACAAATGAAAATTGT TTGAATCTCCGCTTAACATAAATAGCTCGAGGAACTTCCTATACT for the second round. Deletion of *REC8* was similarly performed with GAGGTCATATA CATTGTTTGGCTAGTAGTAGTTGTGTGTGGTGCTTGTTTGTACAATTTTTT AGCAACCACAGGGAACAAAAGCTGGAGCT/ AGCAAACACCTAGCAAAA GAAGTTGCGAAATAAACTGATCACATACAAATCGTTAACCTCATTC-CAAAGGCCTCGAGGAAGTTCCTATACT for both rounds of transformation. The *SAT1* gene was recycled by plating for 100 colonies on SC + maltose to induce expression of the FLP flippase[62] and top spread with 20 μg/mL NAT. Small colonies were screened for loss of *SAT1* by replating to 200 μg/mL NAT. Deletion of *SPO11* or *REC8* was performed in diploid strains prior to integration of the *SAT1* marker at SNP #30[66].

The function of each marker was assessed at each step of strain construction and markers were phased again at the conclusion of strain construction to insure they remained heterozygous and associated with the SNP variant. The *MTL***a**/**a** marked strain and an unmarked *MTL*α/α sibling strain were switched to the opaque state, mated to one another, and the products screened by DNA-staining with Sytox Green followed by flow cytometry to identify tetraploid mating products as described previously[68] and as detailed below.

Reintegration of an intact *SPO11*, *SPO11*(Y65F), or *REC8* was performed in the corresponding Δ*spo11* or Δ*rec8* tetraploid background. The SPO11-YF plasmid was constructed by PCR amplification of the gene in two fragments using oligos GGCGCCCGGGCCCCAAACGTTGATTTCTTGGCC and CAATGGTTGAAAAC TTCCACATCTTGGAAATATATATCACGAATGGTTGTTATTTGAT in one reaction and oligos ATCAAATAACAACCATTCGTGTATATATTTCCAAGA TGTGGAAGTTTTCAACCATTG and GCCGGCCTCGAGTGTGGGAAGCAG TTTCTGAC in a second reaction. The two PCR products were combined by fusion PCR and cloned into the pCR-Blunt II-TOPO vector (Invitrogen) and sequenced to confirm the presence of the Y65F mutation. The resulting vector was treated with enzymes *Apa*I/*Xho*I to release the SPO11(YF) gene and cloned into the corresponding restriction sites in plasmid pSFS2A. The resulting plasmid, pRB589, was linearized with *Bsg*I and used to transform strain CAY7268/CAY7270 to generate strains CAY7701 and CAY7703. A plasmid containing the *REC8* ORF was obtained by amplification of *REC8* using primers 5′-AAGCGGCCGCGGGTTTG GTTTAATTGTGGC-3′ and 5′-GGTGGTGAGCTCCTTGAACACGAACACGG TGA-3′, digesting the resulting PCR product and pSFS2a with *Sac*I and *Not*I, and ligating the two products. The integrity of the *REC8* sequence was assessed by Sanger sequencing of the plasmid. The resulting plasmid, pRB543, was linearized with *Sac*II and transformed into strain CAY7351/CAY7354 to generate strains MAY572/573.

**Assays to determine chromosome instability and recombination.** Cells were induced to undergo CCL by heavily inoculating cells in quadrants on *S. cerevisiae* PRE-SPO medium and culturing at 37 °C as previously described[38]. After 7 days on PRE-SPO medium, cells were scraped from the plates, counted via hemocytometer, and plated to 2-DOG (50 K, 200 K, and two million cells) and to YPD plates (100 cells). The frequency of 2-DOG[R] colonies was determined relative to the number of

colonies present on YPD. Chromosome loss across all strains and conditions appeared to be less efficient across these experiments compared to previous studies[38,39]. 2-DOG plates were also replica plated to synthetic complete defined (SCD) media lacking histidine or uracil to select for *HIS1* or *URA3*, respectively, or to YPD + NAT (200 μg/mL) to select for *SAT1*. The frequency of recombination was identified by the presence or absence of genetic markers (*HIS1/SAT1*) in colonies that were 2-DOG[R] (*gal1*[−]). Recombination frequency was calculated as the number of colonies observed to have recombined between two markers relative to the total number of 2-DOG[R]. Loss of all three markers (*HIS3/URA3/SAT1*) suggested the marked chromosome itself had been lost. All direct comparisons were made between data sets performed in parallel as considerable variation was observed between CCL experiments performed on different days, possibly owing to differences in plate humidity or other environmental conditions.

**Selective genotyping of strains by ddRAD-Seq.** Progeny from either wildtype or Δ*spo11* tetraploid strains that had undergone CCL were plated to 2-DOG and screened by flow cytometry for near-diploid DNA content. Twenty-one colonies from each genotype along with five colonies of each parental genotype were prepared for double digest restriction-site associated (ddRAD-Seq) sequencing essentially as described[69]. In brief, DNA from each sample was digested with *Mbo*I and *Mfe*I restriction endonucleases (New England Biolabs, Ipswich, MA) to produce non-complementary overhangs. P1 and P2 adapters complementary to each overhang, respectively, were ligated to the DNA fragments. P1 adapters encode unique 7 bp barcodes and P2 adapters corresponded to the D701–D712 Illumina adapter set. Adapter-ligated pools of eight samples were gel excised between 200 and 500 bp[69]. The resulting pools were amplified using universal P5 and P7 Illumina amplification primers. Prepared libraries were sequenced on a HiSeq 2500 at the Genewiz Sequencing Facility (South Plainfield, NJ). Approximately two million reads were generated for each sample. Reads were demultiplexed using the bar-code_splitter scripts[70] and quality controlled using FastQC[71]. The ploidy and allelic composition of parental and progeny strains were determined using YMAP[72]. On average, 2169 marker positions were captured across each progeny genome (range: 1458–2513), resulting in an average spacing of ~ 7100 bp between markers.

Recombination breakpoints were determined by comparing the allelic patterns of each progeny to the appropriate parental genotypes. Progeny genotypes that could be inherited from whole parental chromosomes were not considered recombinant. A recombination bin was required to span a minimum of 10 kb as defined by at least four loci that did not match the genotype possible from inheritance of parental chromosomes.

**Flow cytometry of DNA content.** Tetraploid cells passaged or induced to undergo CCL were isolated as single colonies after 7 days. Single colonies were grown overnight in YPD in 96-well plates and prepared for flow cytometry as previously described[68]. In brief, cells were washed and resuspended in 50:50 TE (50 mM Tris, pH 8, 50 mM EDTA) solution. Cells were then treated with 1 mg/mL RNAse A for 4 hours at 37 °C followed by 5 mg/mL Proteinase K treatment at 37 °C for 45 min. Cells were washed with 50:50 TE and resuspended in SybrGreen (1:100 dilution in 50:50 TE) and incubated overnight at 4 °C. Stained cells were washed and resuspended in 50:50 TE and SybrGreen staining data was obtained for 50,000 events for each sample using a BD FACSCanto II. Ten independent known diploid and tetraploid samples served as controls.

**Gam-GFP.** The plasmid pRS11 contains a Gam-GFP cassette under the control of a TetON promoter and allows for visualization of DNA double-strand breaks. *C. albicans* codon-optimized Gam flanked by two *Sal*I restriction sites was synthesized, digested with *Sal*I and cloned into the pNIM1 vector[73] at the *Sal*I site to generate the tetO-Gam-GFP plasmid. For generation of Gam-GFP-marked strains, pRS10 was digested with *Kpn*I/*Sac*II to linearize the plasmid for targeting to the *ADH1* locus and transformants selected on YPD + nourseothricin.

Strains containing the Gam-GFP construct were grown for 24 h on either PRE-SPO or SCD media in the presence or absence of 50 μg/mL doxycycline. Cells were taken from the plates and visualized across 10 fields of view. Nuclear staining was performed with 25 μg/mL 4′,6-diamidino-2-phenylindole.

For western blots, strains containing the Gam-GFP construct were grown for 24 h on YPD in the presence or absence of 50 μg/mL doxycycline. Cell lysates were prepared as previously described[74]. In brief, cells were spun down and resuspended in Thorner buffer (40 mM Tris, pH 8.0, 5% sodium dodecyl sulfate, 8 M urea, and 100 μM EDTA) and transferred to a 2 mL cryovial with 100 μL of 500 micron glass beads. The tubes were heated at 100 °C for 3 min and bead beat for four intervals of 1 min ON/1 min OFF. Western blots were performed with mouse anti-GFP (Sigma, St. Louis, MO) and mouse anti-PSTAIR (Abcam, Cambridge, MA).

**TUNEL assay.** Tetraploid cells grown on PRE-SPO solid medium for 48 h were collected and spheroplasted as previously described[75]. In briefly, 1 mL of cells (OD = 2) were fixed in 4% formaldehyde for 1 h, followed by quenching with 125 mM glycine for 10 min, and then spheroplasted using 5 U of zymolyase according to the manufacturer's instructions (G-Biosciences, St. Louis, MO). Spheroplasting was assessed visually with ~ 90% of the cells appearing as 'ghosts'. Spheroplasted cells were then incubated with TUNEL reagents according to the

manufacturer's instructions (Roche, Basel, Switzerland) to mark DNA double-strand breaks. Cells were visualized by microscopy for fluorescence indicating DNA breaks, and an unlabeled sample served as a negative control.

**Stress resistance**. Overnight cultures were grown in YPD liquid medium at 30 °C. Cell density was determined using $OD_{600}$ and cultures were adjusted to an $OD_{600}$ of 1.0 in 1 mL $ddH_2O$. These dilutions were used as a base for five sequential tenfold dilutions. For each stress condition, 5 µL of each dilution was spotted to the appropriate prewarmed agar plates including a SCD medium plate absent any stressor as a control for growth. Plates were then incubated at 37 °C and imaged at 48 h. Conditions assayed included 100 µg/mL calcofluor white, 2 mM hydroxyurea, 0.01% MMS, and control SCD media. Experiments were performed with a minimum of three biological replicates on separate wildtype, Δspo11, and SPO11-complemented isolates.

**Statistics**. Statistics are provided throughout the manuscript. All two-sample $t$ tests were two-sided comparisons. All measurements were taken from distinct biological samples.

**Reporting summary**. Further information on research design is available in the Nature Research Reporting Summary linked to this article.

## Data availability

Sequence data used for ddRAD-Seq analysis is available at the BioProject database under accession code PRJNA560397. Other data sets generated and/or analyzed during the current study are available in this article and its Supplementary Information files, or from the corresponding author on request.

## Code availability

The code used to generate genotypes for each marker position within the ddRAD-Seq data can be obtained through the YMAP workflow (http://lovelace.cs.umn.edu/Ymap/)[72] or from Dr. Judy Berman's lab upon request.

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

## Acknowledgements

We thank Peter Belenky, Rebecca Shapiro, and Suzanne Noble for gifting plasmids. We thank Judy Berman and Mason Clark for assistance with enumerating recombination assays. This work was supported by National Institutes of Health grants AI081704, AI122011, and AI141893 (to R.J.B.), a PATH award from the Burroughs Wellcome Fund (to R.J.B.), a Research Supplement to Promote Diversity in Health-Related Research award (to M.Z.A.), and a F31 DE023726 (to M.P.H.).

## Author contributions

M.Z.A. and R.J.B. designed the study; M.Z.A. and G.J.T. collected the data; M.Z.A analyzed the data; M.Z.A. and M.P.H. constructed strains used in this study; M.Z.A. and R.J.B. drafted and edited the manuscript.

## Competing interests

The authors declare no competing interests.
