## [Peer Review File · Nature Communications]

Reviewers' comments:

Reviewer #1 (Remarks to the Author):

This exciting manuscript presents evidence for a novel type of parameiotic cycle in the important human fungal pathogen *Candida albicans*. This organism has an unusual version of a parasexual cycle in which MTL α /alpha isolates first undergo homozygosis at MTL, then white to opaque switching, and the resulting diploid specialized mating cells fuse to produce a tetraploid. Under certain conditions these tetraploids undergo something termed concerted chromosome loss to return to the haploid or near haploid state. How this occurs was not understood in detail. Sporulation does not occur, and whether there is meiosis or some vestige of it was largely unknown. The lead author here published a seminar paper some years ago showing that the key meiotic recombinase SPO11 stimulates a low level of recombination during the tetraploid to diploid state, and also presented evidence that it was expressed in mitotic cells, and concluded that the process had been largely converted from a meiotic to a mitotic one.

The studies presented here expand considerably on this foundation.

They explore again the roles of SPO11 and a second meiotic factor REC8 and implicate both in genome stability/instability and in stimulating recombination during concerted chromosome loss.

They implemented a novel method using a protein from Mu phage, Gam, that binds DNA DSB to show that SPO11 generates breaks during CCL. They map in detail recombination events in CCL progeny and show that these are significantly reduced in *spo11* and in *rec8* mutants.

They marshal these findings to propose that the process that is occurring shares features with meiosis, but may be lacking some components, and they suggest appropriately that this could be call "parameiosis", a term that had been suggested by others in reviews.

They cover in the discussion whether this is a derived state, and in fact it most certainly is given that the species is embedded within fungi that undergo complete sexual reproduction including meiosis and sporulation. Thus *Candida* has lost these features.

This model can explain why many meiotic genes remain in *Candida albicans*.

This may provide insights into how meiosis first evolved, if the loss events here mirror gains that may have occurred during evolution.

It seems likely that there may be other eukaryotes that are similarly parameiotic. Are their candidates to propose or discuss?

The authors also discuss recent papers on meiosis like events that can occur in tumor cells, which is an interesting twist.

This is a very clean study, the experiments well conducted and interpreted and the paper is well written.

In my opinion, it should be accepted for Nature Communications.

Minor comments.

1. Figure 2A the panel A images are blurry and should be improved. For panel C, it is very small and hard to see and the quality is low. There is a supplemental figure, S6 that is much higher quality, and the authors should consider how to make the primary figures of higher quality for publication.

2. For the strain designations, KO I think means knock out, which is jargon. The correct genetics term would be deletion.

AB may indicate add back? Again, that would be jargon and the correct term would be complement. There is likely another way to indicate these strains throughout.

3. Is anything known about other meiotic or paramiotic genes and their functions in *Candida*? What about the ortholog of Dmc1 reported many years ago now by Diener and Fink?

4. How many products do the authors think there are from the CCL/paramiosis? A standard meiosis would yield four products in many cases. *Candida lusitanae* only yields dyads so two products of meiosis. For *Candida albicans* do they think it is 1, 2, 3, or 4? Is there any way to address this experimentally? This is a thought question for the future, not a request to do an experiment on this for this paper!

Reviewer #2 (Remarks to the Author):

Candida albicans undergoes an unusual form of sexual reproduction, where tetraploidization is followed by concerted chromosomal loss to re-establish the diploid state. Bennett and colleagues have investigated DNA recombination and the role of meiotic proteins Spo11 and Rec8 during depolyploidization. They found that polyploidization is accompanied by elevated levels of DNA recombination, similar to meiosis, and that this depends in part on Spo11-mediated double-strand break formation, and on Rec8. However, Spo11 and Rec8 have opposite effects on genome stability during depolyploidization: Spo11 promotes chromosomal stability, whereas Rec8 reduces chromosomal stability. Finally, the authors provide intriguing evidence that Spo11 has a function in maintaining genome stability during mitotic growth.

This is a very interesting paper that brings important new insights into the life cycle of *C. albicans* and has study has implications regarding the evolution and degeneration of meiosis. However, there are a number of areas where additional data are needed to strengthen the conclusions.

Major Comments:

1. Figure 1E. It wasn't clear how the recombination frequencies are calculated. The methods section says: 'The frequency of recombination was determined relative to the number of colonies on 2-DOG and in reference to previously determined phasing of the genetic markers.' This could be made more clear. Also, it would be worth explaining at this point the implications of the somewhat surprising similarity in recombination frequencies between the tiny SAT1-URA3 interval and the much larger HIS1-SAT1 interval. This is particularly surprising since the recombination frequencies seem so similar between both intervals whether in standard growth or paramiosis for wild type, or in paramiosis for the SPO11 or REC8 knockouts. Finally, it seems surprising that the difference in recombination frequency between standard growth and paramiosis (2700 to 8200 fold) is much higher than the difference from knocking out SPO11 (24 fold). This could be discussed.

2. The evidence the authors provide to support Spo11-dependent DSB formation depends on the detection of nuclear GAM-GFP, but there are some puzzling aspects of the assay as presented that could be clarified. Specifically, the cells appear to be either positive or negative for signal. Shouldn't it be the case that all cells have a diffuse signal, with nuclear signal only in cells with DSBs? One concern is that some strains express GFP while others fail to do so. This is particularly problematic since genome instability is elevated in the absence of Spo11, hence the cultures assayed may have lost the expression cassette. A western blot analysis against GFP using the cultures in Figure 2E would reveal whether GFP was indeed expressed.

3. Additional controls to validate the DSB responsiveness of the Gam-GFP reporter would also be valuable. Perhaps ideal would be a system with inducible DSBs (inducible restriction enzyme

expression, e.g.), but requiring a system like that is probably outside the scope of what is reasonable for a revision. A straightforward alternative would be ionizing radiation treatment. Also, it should be noted that there is no clear way to interpret how the Gam-GFP signal relates to DSB numbers. If the authors could address this (e.g., quantifying signal vs. radiation dose), that would be helpful. If this cannot be addressed, then it is important to discuss this limitation directly.

4. It would greatly strengthen the manuscript if there were at least one independent assay to verify the presence of DSBs. One would be to perform Southern blot analysis or bulk DNA staining of chromosomes separated on pulsed field gels. Another would be to stain cells or spread chromosomes for typical markers of DSB formation and repair (Rad51 or RPA foci, for example). Another would be to apply TUNEL staining. Without some kind of orthogonal assay, the Gam-GFP results are not really strong enough on their own.

5. Why is the 2-DOG resistant frequency so much higher for the wild type strain in Fig. 2B than in Fig. 1C? Related, the *rec8* knockout strain seems to have a frequency in Fig. 2F that is similar to the wild type frequency in Fig. 1C. Implications of this variability should be explained in the methods since this potentially affects the interpretation of differences between strains. How is this variability controlled for in different experiments?

6. Since the phenotypes of *SPO11* and *REC8* mutants are opposite in the 2-DOGR assay, it would be interesting to know whether this is reflected in the rates of CCL. Was the ploidy of *Rec8* CCL progeny evaluated by FACS?

7. Figure 3: I don't understand panels A and E. Why are the A and B parents different in the two experiments? It is not obvious to me how to read this and how to identify the recombination sites.

8. Figure S2: Why is the *Spo11*-AB ploidy intermediate between WT and KO? This is not discussed. Also, it would be good to have ploidy markers on that graph. Please also provide representative examples of the primary data (FACS plots), not just the summary quantification.

9. In Figure 4A, it would also be interesting to evaluate the ploidy of the *SPO11* mutant tetraploids after culture at 37 °C. Presumably, if these were truly 'mitotic' cultures, the increased frequency of 2-DOGR should not be accompanied by CCL?

10. Throughout: error bars need to be defined, and number of replicates indicated. Instead of bar graphs, it would be much better to provide univariate scatter plots to show the individual value for each determination. See <https://journals.plos.org/plosbiology/article?id=10.1371/journal.pbio.1002128> for a discussion of ways to improve data presentation.

Minor Comments:

11. In all of the frequency bar graphs, please provide tick marks. It is hard to properly interpret the graphs as presented.

12. p. 5 and Fig 1F: quantification of the Gam-GFP positive frequency should be provided at this first mention of the assay, not deferred to the later experiments and Fig S5.

13. Figure 3B, F: Whether different events are assigned the same break point presumably depends on the bin size, which should therefore be indicated.

14. P8: The recombination tract length is discussed in the text but the analysis not shown. It would be better to include that on the Figure too.

15. Figure S5: Where does the significant fraction of GAM-GFP positive cells come from in *Spo11*-

YF? How is this interpreted?

16. P7: 'Trisomies of several smaller chromosomes (4, 5 and 6) were the most common CCL progeny.' Is it possible that this bias may have been introduced because the cells were screened for near diploidy (see Materials)? If not, why do the authors think the short chromosomes should be less prone to chromosomal loss? Based on their results, one may have anticipated the opposite, since short chromosomes recombined less. But perhaps there is no obvious prediction? Note that Figure S7 doesn't clearly convey the effect of chromosome size.

Reviewer #3 (Remarks to the Author):

This is an important paper that appears to have revealed a novel process that shares properties with meiosis, but only results in partial reduction of ploidy. Importantly, Spo11 is required for the high level of recombinants seen in the system and the meiotic cohesion RecA is required for the observed loss of heterozygosity (uncovering of a recessive gal marker). The work is certain to be of broad interest to the readers of Nature Communication and I therefore recommend it be published provided the comments below are addressed in a satisfactory manner.

P4. What the authors are calling *S. cerevisiae* PRE-SPO medium is unfamiliar to me. The budding yeast PRE-SPO medium I am familiar with uses either acetate for a carbon source (YPA or SPS). Is the recipe given on P. 13 correct? A citation of an *S. cerevisiae* paper that employs this type of medium should be given. It would also be helpful if the authors are aware of what component of their PRE-SPO medium differs from normal mitotic growth medium and is therefore thought to be responsible for inducing CCL. Is it basically a partial starvation regimen?

P.4 is It would be helpful to mention that SAT1 confers resistance to NAT. This mentioned in the methods, but would be helpful to mention in the Figure 1 legend as well.

P.5 How does the Gam-tag work? What is being scored is GFP positive nuclei, but the protein should be expressed in cells both with and without DSBs so one expects that all nuclei will be GFP positive, unless binding to DNA ends is required for protein stability OR the nuclei are extracted to remove soluble protein prior to scoring.

Figure 2. The designation SPO11-AB is not defined. I am assuming AB means "add-back" but this does not appear to be mentioned anywhere.

P 8. What statistical analysis was done to determine that the clusters of recombination events observed reflect a higher rate of recombination in those regions as opposed to random? Same question regarding the claim that the spo11 mutant is specifically defective in internal events.

P 8. I am. Not clear on how the global detection of recombination events works. I presume the diploids used to construct the tetraploid have diverged. Are the two chromosomes of the parent diploids also diverged? Specifically, exactly how many markers are scored? What is the marker density? How does the marker distribution effect the ability to detect events in different regions of the genome? Given that there are 4 homologs, is there evidence for a preference in recombination between the two homologs from the same parent diploid?

P 8. I am confused by the discussion of recombination tract lengths. Have the authors considered the possibility that they are looking at reciprocal crossing over for cases where the recombination tract goes to the end of the chromosome? That seems far more likely to me than BIR which is a gene conversion process that is rare in meiosis compared to crossing over. In general the use of the term "tract length" is properly reserved for gene conversions. In this system one could argue the shorter tracts are likely gene conversions, but it is not clear to me how the authors know the

terminal events are conversions. Was there an effort to detect reciprocal events?

P. 9 Authors might comment on why they think the defect in recombination in *spo11Δ* is not as severe as that of *spo11-YF*. One possibility is that the catalytically inactive protein is interfering with an alternative pathway for forming recombination events. However, one could be concerned that there is some unknown source of variability. Was this experiment repeated? If not, it should be.

P. 10 , Figure 4. The results showing *spo11* mutants are sensitive to MMS are quite thin. Given the extraordinary nature of the claim regarding an unexpected function for Spo11 in damage repair, this experiment should be performed in triplicate. It would be even better if there were individual isolates of the *spo11* mutant to be sure that the strain is not defective in repair for some other reason.

P. 11. The impact of the *rec8* mutation in CCR could reflect the possibility that uncovering recessive markers requires reductional segregation promoted by Rec8. This would explain why *spo11* mutants increase instability if Rec8 is promoting reductional segregation without recombination to stabilize proper bi-polar attachment to the spindle. In this view, *spo11* mutants display a big increase in loss of heterozygosity because of Rec8+- driven reductional segregation. This view predicts that the high stability of the *rec8* single mutant will be epistatic to the low stability of the *spo11* single mutant. I won't ask for the epistasis experiment for this paper, but I recommend the authors consider looking at a *rec8 spo11* double in the future as it would clarify the mechanism of Rec8 driven instability and also suggest that Spo11 formed reciprocal recombinants can undergo proper reductional segregation during CCL.

Figure S4. Key should identify KO as *spo11-KO*

Figure S7. How many clones were tested for aneuploidy in each case? i.e what is the sample size? State in Figure legend

Figure S10. Are the 4 strains shown above the heading "SPO11 genotype" tetraploid? If so, does SPO11-WT differ from the results labelled as tetraploid? See comment below regarding identifying the specific strains used.

Figure S11. As for the MMS experiment, measuring damage sensitivity with a single dilution series is dangerous. The negative results presented in this figure should be shown in triplicate.

I think it would be wise if the authors included specific strain designations in the figure legend so that there is no confusion in the future regarding which strain was used in each experiment. It is also customary in the budding yeast community to include a strain table in the supplementary materials with the complete genotype of each strain listed.

We would like to thank the 3 reviewers for their detailed comments on the manuscript. We agree with many of the comments and have performed additional experiments and editing of the text to address them. We hope the changes made to the manuscript address the concerns of the reviewers.

Reviewers' comments:

Reviewer #1 (Remarks to the Author):

This exciting manuscript presents evidence for a novel type of parametiotic cycle in the important human fungal pathogen *Candida albicans*. This organism has an unusual version of a parasexual cycle in which MTL α isolates first undergo homozygosity at MTL, then white to opaque switching, and the resulting diploid specialized mating cells fuse to produce a tetraploid. Under certain conditions these tetraploids undergo something termed concerted chromosome loss to return to the haploid or near haploid state. How this occurs was not understood in detail. Sporulation does not occur, and whether there is meiosis or some vestige of it was largely unknown. The lead author here published a seminar paper some years ago showing that the key meiotic recombinase SPO11 stimulates a low level of recombination during the tetraploid to diploid state, and also presented evidence that it was expressed in mitotic cells, and concluded that the process had been largely converted from a meiotic to a mitotic one.

The studies presented here expand considerably on this foundation.

They explore again the roles of SPO11 and a second meiotic factor REC8 and implicate both in genome stability/instability and in stimulating recombination during concerted chromosome loss.

They implemented a novel method using a protein from Mu phage, Gam, that binds DNA DSB to show that SPO11 generates breaks during CCL. They map in detail recombination events in CCL progeny and show that these are significantly reduced in *spo11* and in *rec8* mutants.

They marshal these findings to propose that the process that is occurring shares features with meiosis, but may be lacking some components, and they suggest appropriately that this could be called "parametiosis", a term that had been suggested by others in reviews.

They cover in the discussion whether this is a derived state, and in fact it most certainly is given that the species is embedded within fungi that undergo complete sexual reproduction including meiosis and sporulation. Thus *Candida* has lost these features.

This model can explain why many meiotic genes remain in *Candida albicans*.

This may provide insights into how meiosis first evolved, if the loss events here mirror gains that may have occurred during evolution.

It seems likely that there may be other eukaryotes that are similarly parametiotic. Are there candidates to propose or discuss? The authors also discuss recent papers on meiosis like events that can occur in tumor cells, which is an interesting twist.

It is now recognized that a variety of mechanisms can support ploidy cycling across a wide range of organisms. We previously mentioned other examples of depolyploidization such as those that accompany cancer progression, where a role for 'meiosis' genes has also been implicated. We

have now expanded the Discussion to include connections with other potential non-meiotic processes including those in fungi, parasites and mammalian cells.

This is a very clean study, the experiments well conducted and interpreted and the paper is well written. In my opinion, it should be accepted for Nature Communications.

We thank Reviewer 1 for their support of the work and their interest in the study.

Minor comments.

1. Figure 2A the panel A images are blurry and should be improved. For panel C, it is very small and hard to see and the quality is low. There is a supplemental figure, S6 that is much higher quality, and the authors should consider how to make the primary figures of higher quality for publication.

A number of figures have been resized and reformatted to increase clarity and we thank the reviewer for noting this.

2. For the strain designations, KO I think means knock out, which is jargon. The correct genetics term would be deletion.

AB may indicate add back? Again, that would be jargon and the correct term would be complement. There is likely another way to indicate these strains throughout.

The reviewer is correct that 'KO' refers to gene knockouts and 'AB' to gene addbacks. We have changed these terms throughout the figures and text per the reviewer's suggestion. The corresponding new terms are " $\Delta spo11$ " and " $\Delta +SPO11(WT)$ ".

3. Is anything known about other meiotic or parameiotic genes and their functions in *Candida*? What about the ortholog of Dmc1 reported many years ago now by Diener and Fink?

We agree this is important to discuss and have included additional information regarding the ortholog of *DMC1* in *C. albicans* (named *DLH1*) on lines 234-236. We are currently unaware of any additional meiosis-specific genes that have been specifically targeted for investigation during the parasexual cycle. A comprehensive list of meiosis-specific genes that are present or absent from the *C. albicans* genome is described in Tzung *et al*, *PNAS*, 2001 (PMID 11248064) which is referenced.

4. How many products do the authors think there are from the CCL/parameiosis? A standard meiosis would yield four products in many cases. *Candida lusitanae* only yields dyads so two products of meiosis. For *Candida albicans* do they think it is 1, 2, 3, or 4? Is there any way to address this experimentally? This is a thought question for the future, not a request to do an experiment on this for this paper!

We appreciate the interest in this question. Currently, we do not know how many products result from parasex. We envisage that a process such as unequal nuclear division is responsible for the observed ploidy reduction, but such decreases in DNA content are not necessary to be isolated from the pre-sporulation plates. It is possible that a single tetraploid cell could give rise to one viable progeny cell or that the tetraploid cell produces multiple progeny with a reduced

ploidy and this process is repeated to produce recombinant parasexual progeny cells. Ongoing efforts are attempting to look at the sequence of events during parasex.

Reviewer #2 (Remarks to the Author):

Candida albicans undergoes an unusual form of sexual reproduction, where tetraploidization is followed by concerted chromosomal loss to re-establish the diploid state. Bennett and colleagues have investigated DNA recombination and the role of meiotic proteins Spo11 and Rec8 during depolyploidization. They found that polyploidization is accompanied by elevated levels of DNA recombination, similar to meiosis, and that this depends in part on Spo11-mediated double-strand break formation, and on Rec8. However, Spo11 and Rec8 have opposite effects on genome stability during depolyploidization: Spo11 promotes chromosomal stability, whereas Rec8 reduces chromosomal stability. Finally, the authors provide intriguing evidence that Spo11 has a function in maintaining genome stability during mitotic growth.

This is a very interesting paper that brings important new insights into the life cycle of *C. albicans* and has study has implications regarding the evolution and degeneration of meiosis. However, there are a number of areas where additional data are needed to strengthen the conclusions.

Major Comments:

1. Figure 1E. It wasn't clear how the recombination frequencies are calculated. The methods section says: 'The frequency of recombination was determined relative to the number of colonies on 2-DOG and in reference to previously determined phasing of the genetic markers.' This could be made more clear. Also, it would be worth explaining at this point the implications of the somewhat surprising similarity in recombination frequencies between the tiny SAT1-URA3 interval and the much larger HIS1-SAT1 interval. This is particularly surprising since the recombination frequencies seem so similar between both intervals whether in standard growth or parameiosis for wild type, or in parameiosis for the SPO11 or REC8 knockouts. Finally, it seems surprising that the difference in recombination frequency between standard growth and parameiosis (2700 to 8200 fold) is much higher than the difference from knocking out SPO11 (24 fold). This could be discussed.

We have clarified the section describing how recombination frequencies were calculated to provide more information (lines 357-364).

The similarity in recombination frequency between the two tested intervals also surprised us given their difference in size. There are a number of reasons that could explain this result as described in detail here:

<https://www.sciencedirect.com/science/article/pii/S0959437X02003581>. These reasons can include GC content, presence of repetitive DNA, heterochromatin structure, and proximity to centromeres, among other factors. It is unclear which of these are operating here but recombination within the *HIS1 – SAT1* interval is not expected to cross the centromere and, therefore, represents two shorter chromosomal segments. These effects would be expected to

behave similarly across the different conditions as the underlying genomic features remain constant in the different genotypes and conditions.

It is true that a larger difference in recombination rates is observed between mitotic and parasexual cells during growth on solid media than between WT and $\Delta spo11$ cells during parasex. We interpret this as being indicative of the massive effect that PRE-SPO medium has on *C. albicans* cells relative to YPD medium. In addition, as noted in the Discussion, the data suggests that additional factors contribute to parasexual recombination in addition to Spo11, as recombination rates are only partially reduced in cells lacking this protein. This is now further highlighted in the revised text (lines 217-221).

2. The evidence the authors provide to support Spo11-dependent DSB formation depends on the detection of nuclear GAM-GFP, but there are some puzzling aspects of the assay as presented that could be clarified. Specifically, the cells appear to be either positive or negative for signal. Shouldn't it be the case that all cells have a diffuse signal, with nuclear signal only in cells with DSBs? One concern is that some strains express GFP while others fail to do so. This is particularly problematic since genome instability is elevated in the absence of Spo11, hence the cultures assayed may have lost the expression cassette. A western blot analysis against GFP using the cultures in Figure 2E would reveal whether GFP was indeed expressed.

The reviewer is correct that diffuse GFP signal should be observed throughout the cell prior to accumulation within the nucleus. We have provided a more consistent image to insure this is clearer than in the original version. Additionally, a Western Blot of GAM-GFP expression was now performed to demonstrate its regulation by doxycycline and this establishes similar expression levels across genotypes (see Figure S3). For Western blots, cells were grown identically to the conditions used for microscopy - 24 hours of growth on PRE-SPO medium in the presence or absence of doxycycline.

3. Additional controls to validate the DSB responsiveness of the Gam-GFP reporter would also be valuable. Perhaps ideal would be a system with inducible DSBs (inducible restriction enzyme expression, e.g.), but requiring a system like that is probably outside the scope of what is reasonable for a revision. A straightforward alternative would be ionizing radiation treatment. Also, it should be noted that there is no clear way to interpret how the Gam-GFP signal relates to DSB numbers. If the authors could address this (e.g., quantifying signal vs. radiation dose), that would be helpful. If this cannot be addressed, then it is important to discuss this limitation directly.

We have performed the suggested experiment by exposing cells to various doses of gamma-irradiation to assess Gam-GFP signal in response to DNA damage. Gam-GFP signal accumulated in the nucleus following irradiation for all genotypic backgrounds. While it is challenging to quantify the GFP signal in the nucleus, there was a perceived increase in brightness for cells exposed to higher doses of ionizing radiation (25 grays vs. 10 grays). These experiments are presented in Figure S4.

4. It would greatly strengthen the manuscript if there were at least one independent assay to verify the presence of DSBs. One would be to perform Southern blot analysis or bulk DNA staining of chromosomes separated on pulsed field gels. Another would be to stain cells or spread chromosomes for typical markers of DSB formation and repair (Rad51 or RPA foci, for example). Another would be to

apply TUNEL staining. Without some kind of orthogonal assay, the Gam-GFP results are not really strong enough on their own.

We agree that an independent assay for DSB formation would significantly strengthen the results. To achieve this, cells were exposed to CCL-inducing conditions (PRE-SPO solid agar medium at 37°C) and assessed for DSB formation by TUNEL staining, as per the reviewer's suggestion. TUNEL staining was evident in a large fraction of cells that expressed *SPO11*, yet was less frequent in cells lacking *SPO11*. This data is now provided in Figure S6 and strongly supports the phenotypes generated by GAM-GFP.

5. Why is the 2-DOG resistant frequency so much higher for the wild type strain in Fig. 2B than in Fig. 1C? Related, the *rec8* knockout strain seems to have a frequency in Fig. 2F that is similar to the wild type frequency in Fig. 1C. Implications of this variability should be explained in the methods since this potentially affects the interpretation of differences between strains. How is this variability controlled for in different experiments?

The reviewer is correct to note differences in 2-DOG^R rates between different experiments. This is due-to-day variation in the assays used to induce chromosome loss. This variability is controlled for by running all strains and all replicates for each experimental in parallel to be able to accurately compare the impact of individual genes or conditions (temp/media/ploidy) on chromosome loss and recombination rates. We now highlight this in the text (lines 362-364).

6. Since the phenotypes of *SPO11* and *REC8* mutants are opposite in the 2-DOG^R assay, it would be interesting to know whether this is reflected in the rates of CCL. Was the ploidy of *Rec8* CCL progeny evaluated by FACS?

We have now included the ploidy of cells with different *REC8* genotypes as part of the manuscript in Figure S10. This data is consistent with the frequency of 2-DOG^R colonies observed.

7. Figure 3: I don't understand panels A and E. Why are the A and B parents different in the two experiments? It is not obvious to me how to read this and how to identify the recombination sites.

The differences in parental genotypes A/B in panel E of Figure 3 reflect the way that these strains were constructed. As noted in Figure S1, deletion of *SPO11* required additional construction steps beyond those required for the WT strains used to build the tetraploid for CCL. These strain construction steps can induce a certain amount of chromosomal change as has been observed previously in *C. albicans* (see <https://www.ncbi.nlm.nih.gov/pubmed/22384363> and <https://www.ncbi.nlm.nih.gov/pubmed/27206717>). To aid the reader in visualization of recombination sites within A and E, the sites of recombination have been marked with an arrow. Note that the color changes marked in the $\Delta spo11$ progeny match the breakpoints present in the parents, indicating inheritance of a chromosome homolog that had undergone loss of heterozygosity (LOH) prior to CCL. In contrast, color patterns present in the WT progeny are not found in the parental genotypes, highlighting recombination breakpoints that occurred during the CCL process.

8. Figure S2: Why is the Spo11-AB ploidy intermediate between WT and KO? This is not discussed. Also, it would be good to have ploidy markers on that graph. Please also provide representative examples of the primary data (FACS plots), not just the summary quantification.

We posit that the intermediate ploidy phenotype in the *SPO11*-complemented strain is due, in part, to the presence of only a single *SPO11* allele within this strain compared to four copies in the WT strain. (In addition, if the chromosome encoding the *SPO11* locus is lost during CCL, the strain will now genotypically resemble a $\Delta spo11$ strain). We hypothesize this is why most data for the *SPO11*-complemented strain resembles the WT and is statistically equivalent while a few data points are lower in ploidy and may be the result of loss of the chromosome homolog encoding the intact *SPO11* allele or reduced *SPO11* gene dosage effects (4 copies in the WT and 1 copy in the complemented strain). This point has been added (lines 115-116). Primary flow plots have been added in panel A of Supplemental Figure 2 to display flow cytometry plots of diploid, aneuploid, and tetraploid cell populations following parasexual induction.

9. In Figure 4A, it would also be interesting to evaluate the ploidy of the *SPO11* mutant tetraploids after culture at 37 °C. Presumably, if these were truly 'mitotic' cultures, the increased frequency of 2-DOGR should not be accompanied by CCL?

We would like to note that *SPO11* mutant strain data is provided in Figure 4 for growth at 37°C. As noted in the text, there is an increased 2-DOGR^R rate for tetraploids grown at 37°C but not for either diploids at 37°C or tetraploids grown at 30°C. We propose that this is associated with increased chromosome loss and movement of the tetraploid to a reduced ploidy state under these conditions. A previous study revealed similar chromosome instability of tetraploid cells compared to their diploid counterparts under extended passaging (<https://www.ncbi.nlm.nih.gov/pubmed/25991822>).

10. Throughout: error bars need to be defined, and number of replicates indicated. Instead of bar graphs, it would be much better to provide univariate scatter plots to show the individual value for each determination. See <https://journals.plos.org/plosbiology/article?id=10.1371/journal.pbio.1002128> for a discussion of ways to improve data presentation.

We thank the reviewer for pointing out these omissions. We have added definitions to the error bars and the number of replicates throughout the figures. We have also revised how the data is displayed. All data points are plotted for recombination and chromosome loss assays as well as for frequency of recombination in Figure 3. The violin plots represent chromosome loss assays and boxplots denote recombination assays.

Minor Comments:

11. In all of the frequency bar graphs, please provide tick marks. It is hard to properly interpret the graphs as presented.

Tick marks have been added in the figures as suggested.

12. p. 5 and Fig 1F: quantification of the Gam-GFP positive frequency should be provided at this first mention of the assay, not deferred to the later experiments and Fig S5.

This has been edited per the reviewer's suggestion.

13. Figure 3B, F: Whether different events are assigned the same break point presumably depends on the bin size, which should therefore be indicated.

This information has been provided per the reviewer's suggestion in the Methods (line 382-386).

14. P8: The recombination tract length is discussed in the text but the analysis not shown. It would be better to include that on the Figure too.

Recombination tract length is provided in panel D of Figure 3 for the WT strain. The $\Delta spo11$ strain contains very few recombination breakpoints so that a comparison between the two would be difficult to interpret based on the number of events observed in the current study.

15. Figure S5: Where does the significant fraction of GAM-GFP positive cells come from in Spo11-YF? How is this interpreted?

The author is correct that there is a larger fraction of Gam-GFP positive cells in the SPO11-YF strain than in the $\Delta spo11$ mutant (although less than in cells complemented with the WT SPO11 allele) and this is now indicated in the main text. It is not obvious how the mutant Spo11 is able to support a limited amount of DSB formation. It is possible that Spo11-YF may bind DNA but is unable to cleave since it is lacking the catalytic residue and blockage of DNA polymerase may increase DSB formation, or the mutant protein may bind to DSBs formed by other mechanisms and interfere with their processing. At this point, we would prefer not to speculate on this mechanism without further studies given that the effect size is small.

16. P7: 'Trisomies of several smaller chromosomes (4, 5 and 6) were the most common CCL progeny.' Is it possible that this bias may have been introduced because the cells were screened for near diploidy (see Materials)? If not, why do the authors think the short chromosomes should be less prone to chromosomal loss? Based on their results, one may have anticipated the opposite, since short chromosomes recombined less. But perhaps there is no obvious prediction? Note that Figure S7 doesn't clearly convey the effect of chromosome size.

Cells for ddRAD-Seq were selected based on a DNA content expected to be between diploid and triploid by flow cytometry. It is difficult to distinguish between 2N and 2N+1 based on a slight shift in the total SYTOX Green staining of cells within the population (see PMID 28752816 under heading 'Molecular Detection of Ploidy and Aneuploidy'). While it is possible that a bias in selection exists, there was no systematic bias for smaller chromosomes. Furthermore, previous reports of chromosome stability also highlight preferential retention of smaller chromosomes during aneuploid formation (PMID 18927630, 18462019, and 22804579) even when selection is less strongly applied to the population.

Reviewer #3 (Remarks to the Author):

This is an important paper that appears to have revealed a novel process that shares properties with meiosis, but only results in partial reduction of ploidy. Importantly, Spo11 is required for the high level of recombinants seen in the system and the meiotic cohesion RecA is required for the observed loss of heterozygosity (uncovering of a recessive gal marker). The work is certain to be of broad interest to the

readers of Nature Communication and I therefore recommend it be published provided the comments below are addressed in a satisfactory manner.

P4. What the authors are calling *S. cerevisiae* PRE-SPO medium is unfamiliar to me. The budding yeast PRE-SPO medium I am familiar with uses either acetate for a carbon source (YPA or SPS). Is the recipe given on P. 13 correct? A citation of an *S. cerevisiae* paper that employs this type of medium should be given. It would also be helpful if the authors are aware of what component of their PRE-SPO medium differs from normal mitotic growth medium and is therefore thought to be responsible for inducing CCL. Is it basically a partial starvation regimen?

The recipe for PRE-SPO medium is taken from a previous publication (PMID 16535013). The recipe has been consistently used by the Bennett lab and was originally obtained from the lab of Ira Herskowitz (UCSF). This reference has been added in the Methods section.

P.4 is It would be helpful to mention that SAT1 confers resistance to NAT. This mentioned in the methods, but would be helpful to mention in the Figure 1 legend as well.

This has been edited per the reviewer's suggestion (line 604-605).

P.5 How does the Gam-tag work? What is being scored is GFP positive nuclei, but the protein should be expressed in cells both with and without DSBs so one expects that all nuclei will be GFP positive, unless binding to DNA ends is required for protein stability OR the nuclei are extracted to remove soluble protein prior to scoring.

The Gam-GFP construct is under the control of a Tet-ON promoter. In the absence of doxycycline there is no expression of the Gam-GFP protein whereas it is produced when doxycycline is present. In the presence of DNA damage, Gam-GFP accumulates in the nucleus where it binds to DSBs. Thus, there is a weak diffuse signal throughout the cell in the absence of DSBs but a concentrated signal in the nucleus following DNA damage (see Figure 3). We have added additional data (see also Reviewer #2 comments) to demonstrate it is present only when doxycycline is added to cells (Figure S3) and accumulates in the nucleus in response to DNA damage (Figure S4), as well as provided TUNEL data as an alternative approach (see Figure S6).

Figure 2. The designation SPO11-AB is not defined. I am assuming AB means "add-back" but this does not appear to be mentioned anywhere.

The use of 'AB' as add-back has been replaced with ' Δ +SPO11(WT)' throughout, to also address Reviewer #2 comments.

P 8. What statistical analysis was done to determine that the clusters of recombination events observed reflect a higher rate of recombination in those regions as opposed to random? Same question regarding the claim that the *spo11* mutant is specifically defective in internal events.

An independent two-sample t-test had originally been applied to this comparison. We have updated the comparison to be assessed by a Mann-Whitney U test due to some skew in the data. The comparison is significantly different using the non-parametric test.

We attempted to test the effect on Spo11 recombination tract length but have relatively few $\Delta spo11$ data points to use in this comparison (4 vs. 54 for $\Delta spo11$ and WT, respectively). A

Mann-Whitney U test failed to support a difference between the two, $p = 0.48$. We have edited the text to reflect this result (lines 168-171).

P 8. I am. Not clear on how the global detection of recombination events works. I presume the diploids used to construct the tetraploid have diverged. Are the two chromosomes of the parent diploids also diverged? Specifically, exactly how many markers are scored? What is the marker density? How does the marker distribution effect the ability to detect events in different regions of the genome? Given that there are 4 homologs, is there evidence for a preference in recombination between the two homologs from the same parent diploid?

We agree with the reviewer that this additional information should have been provided. On average, 2169 marker positions were captured across each progeny genome (range: 1458-2513), resulting in an average spacing of ~7100 bp between markers. We do not expect marker distribution to affect the main comparison here between wildtype and $\Delta spo11$ cells as they are both derived from the same parental SC5314 strain. The enzymes used for ddRAD-Seq will produce a similar marker profile across all genomes that is reflected in the data, as the majority of genomes analyzed contained a similar number of marker positions (median of 2226 sites).

Testing for evidence of recombination between two homologs from the same parent would be challenging as both strains used in each mating (marked and unmarked chromosome strains) are derived from identical progenitors. That being said, there are cases where genotypes diverged due to LOH as seen in Figure 2. Identification of preferential recombination would be best performed in regions where each parental genotype was homozygous for opposing alleles (aa vs. bb). No such position exist in the WT parental genotypes and only one arm of Chr5 (Chr5R) has this desired configuration in the $\Delta spo11$ where there is no evidence for recombination. Thus, we cannot look for evidence of biases in recombination from the two chromosome homologs from the same parent with confidence.

P 8. I am confused by the discussion of recombination tract lengths. Have the authors considered the possibility that they are looking at reciprocal crossing over for cases where the recombination tract goes to the end of the chromosome? That seems far more likely to me than BIR which is a gene conversion process that is rare in meiosis compared to crossing over. In general the use of the term “tract length” is properly reserved for gene conversions. In this system one could argue the shorter tracts are likely gene conversions, but it is not clear to me how the authors know the terminal events are conversions. Was there an effort to detect reciprocal events?

We agree with the reviewer that we cannot distinguish between a reciprocal crossing over and a BIR replication event as we are unable to track all of the products from parasexual recombination events. One study used half-sector analysis to show that most mitotic recombination events in *C. albicans* that extend to the ends of the chromosomes occur via BIR (PMID 21791579). However, we have revised the text to indicate that these tracts could be via BIR or reciprocal crossovers (Line 151-154). Again, we would need to recover the recombinant products from all events to distinguish these possibilities.

P. 9 Authors might comment on why they think the defect in recombination in $spo11\Delta$ is not as severe as that of $spo11-YF$. One possibility is that the catalytically inactive protein is interfering with an

alternative pathway for forming recombination events. However, one could be concerned that there is some unknown source of variability. Was this experiment repeated? If not, it should be.

We agree with the reviewer that this result was unexpected and yet is reproducible. These experiments were repeated on multiple occasions (at both institutions) with consistent outcomes. It is certainly possible that there is some interference with an alternate process that promotes DNA breaks or recombination, although what that mechanism is lies beyond the scope of this work. These data demonstrated that the catalytic function of Spo11 contributes to overall levels of recombination.

P. 10 , Figure 4. The results showing *spo11* mutants are sensitive to MMS are quite thin. Given the extraordinary nature of the claim regarding an unexpected function for Spo11 in damage repair, this experiment should be performed in triplicate. It would be even better if there were individual isolates of the *spo11* mutant to be sure that the strain is not defective in repair for some other reason.

We apologize for any confusion. These experiments were performed on multiple occasions with a minimum of three biological replicates. Similar results were observed in experiments performed at both institutions. It is also worth noting that multiple independent $\Delta spo11$ isolates were tested with similar results. The data provided in the figure are a representative example of the effect. We have updated the methods to reflect the above information.

Images for additional replicates are attached here:

P. 11. The impact of the *rec8* mutation in CCR could reflect the possibility that uncovering recessive markers requires reductional segregation promoted by Rec8. This would explain why *spo11* mutants increase instability if Rec8 is promoting reductional segregation without recombination to stabilize proper bi-polar attachment to the spindle. In this view, *spo11* mutants display a big increase in loss of heterozygosity because of Rec8+- driven reductional segregation. This view predicts that the high

stability of the *rec8* single mutant will be epistatic to the low stability of the *spo11* single mutant. I won't ask for the epistasis experiment for this paper, but I recommend the authors consider looking at a *rec8 spo11* double in the future as it would clarify the mechanism of Rec8 driven instability and also suggest that Spo11 formed reciprocal recombinants can undergo proper reductional segregation during CCL.

We are grateful to the reviewer for this insight as we had not considered this possibility. We certainly agree with the need to examine a *spo11 rec8* double mutant in the future.

Figure S4. Key should identify KO as *spo11-KO*

This has been edited as "*Δspo11*" throughout the figures and text.

Figure S7. How many clones were tested for aneuploidy in each case? i.e what is the sample size? State in Figure legend

This information has been included per the reviewer's suggestion.

Figure S10. Are the 4 strains shown above the heading "SPO11 genotype" tetraploid? If so, does SPO11-WT differ from the results labelled as tetraploid? See comment below regarding identifying the specific strains used.

The strains under the heading "SPO11 genotype" all began as tetraploid cells prior to passaging. The WT under that heading is not different from the tetraploid population using a Mann-Whitney U test. The only significantly different genotype is the 'YF' mutant, consistent with a strong chromosome loss phenotype observed in the 2-DOG^R assay (Figure 4).

Figure S11. As for the MMS experiment, measuring damage sensitivity with a single dilution series is dangerous. The negative results presented in this figure should be shown in triplicate.

We have performed these experiments in triplicate, as suggested, and there is a consistent defect in the mutant background. Images of the experiment are attached above.

I think it would be wise if the authors included specific strain designations in the figure legend so that there is no confusion in the future regarding which strain was used in each experiment. It is also customary in the budding yeast community to include a strain table in the supplementary materials with the complete genotype of each strain listed.

A strain table is now included. Due to the large number of strains used we did not opt to include strain numbers in the figure legends.

REVIEWERS' COMMENTS:

Reviewer #3 (Remarks to the Author):

I am satisfied with the responses to my comments.